# Stabilization of Palygorskite Aqueous Suspensions Using Bio-Based and Synthetic Polyelectrolytes

**DOI:** 10.3390/polym13010129

**Published:** 2020-12-30

**Authors:** Eduardo Ferraz, Luís Alves, Pedro Sanguino, Julio Santarén, Maria G. Rasteiro, José A. F. Gamelas

**Affiliations:** 1Techn&Art, Quinta do Contador, Polytechnic Institute of Tomar, Estrada da Serra, PT-2300-313 Tomar, Portugal; ejmoferraz@ipt.pt; 2Geobiotec, Geosciences Department, University of Aveiro, Campus Universitário de Santiago, PT-3810-193 Aveiro, Portugal; 3CIEPQPF, Department of Chemical Engineering, University of Coimbra, Rua Sílvio Lima, Pólo II, PT-3030-790 Coimbra, Portugal; mgr@eq.uc.pt; 4CEMMPRE, Mechanical Engineering Department, University of Coimbra, Rua Luís Reis Santos, Pólo II, PT-3030-788 Coimbra, Portugal; pesang@sapo.pt; 5TOLSA, SA, Research & Technology for New Businesses, Ctra. de Madrid a Rivas Jarama, 35, ES-28031 Madrid, Spain; jsantaren@tolsa.com

**Keywords:** attapulgite, fibrous clay, dispersion, organic–inorganic hybrid, bionanocomposite

## Abstract

Palygorskite is a natural fibrous clay mineral that can be used in several applications, for which colloidal stability in aqueous suspensions is a key point to improve its performance. In this study, methods of magnetic stirring, high-speed shearing, and ultrasonication, as well as different chemical dispersants, combined with these methods, namely carboxymethylcellulose, alginate, polyphosphate, and polyacrylate, were used to improve the dispersibility and the formation of stable suspensions of palygorskite in different conditions of pH. The stability and particle size of suspensions with a low concentration of palygorskite were evaluated by visual inspection, optical and electron microscopy, dynamic light scattering, and zeta potential measurements. Moreover, the palygorskite used in this work was initially characterized for its mineralogical, chemical, physical, and morphological properties. It was found that more stable suspensions were produced with ultrasonication compared to the other two physical treatments, with magnetic stirring being inefficient in all tested cases, and for higher pH values (pH of 12 and pH of 8, the natural pH of the clay) when compared to lower pH values (pH of 3). Remarkably, combined with ultrasonication, carboxymethylcellulose or in a lesser extent polyphosphate at near neutral pH allowed for the disaggregation of crystal bundles of palygorskite into individualized crystals. These results may be helpful to optimize the performance of palygorskite in several domains where it is applied.

## 1. Introduction

Palygorskite is a natural clay mineral with fibrous morphology and an ideal formula (Mg,Al,Fe)_5_Si_8_O_20_(OH)_2_(OH_2_)_4_·4H_2_O, whose structure is based on blocks alternating with cavities (tunnels). The structural blocks contain two tetrahedral silica sheets, with the tetrahedra inverted from one sheet to the other, sandwiching a central octahedral sheet of metal oxide-hydroxide, where the metal ions are mainly magnesium and aluminum, with iron in a lesser extent. In the octahedral sheet, both hydroxyl ions and water molecules are coordinated to metal centers, while the remaining water molecules occupy positions inside the tunnels. The presence of tunnels with cross-sectional dimensions of 0.64 × 0.37 nm^2^ [1] enables the retention of small organic molecules. The bonding between nonshared oxygens from the tetrahedral silica sheets on the external surface of the particles and hydrogen provides a high density of silanol groups on the particle surface [2].

Several micro- and nanoscale applications for the palygorskite—namely in sorption processes; catalytic, rheological, and environmental; in organo-mineral hybrid composites, films, membranes, and bioplastics; for drug delivery; for tissue engineering; as a source of supported carbonaceous materials; and as a component of sensor devices or bioreactors—were collected by Galán (1996), Ruiz-Hitzky et al. (2013) and Wang and Wang (2016) [3,4,5].

The acicular individual particles of palygorskite have a propensity to aggregate in water to form bundles and large aggregates, mainly through hydrogen bonding and van der Waals interactions [3], which makes the preparation of stable colloidal dispersions of this fibrous mineral difficult. In addition, fibrous minerals, because of their crystallochemical structure, cannot be delaminated/exfoliated as plate-like minerals. The dispersion of palygorskite thus represents an important issue to be solved before the processing of this mineral, for which several strategies have been proposed. Wang and Wang (2016) compiled recent progresses of methodologies used to improve dispersion of palygorskite and extend its application [3]. The authors divided the disaggregation methods between dry and wet methods. Dry methods include ball grinding/stone milling, extrusion, or ion-irradiation, while wet methods (applied in a suspension of the mineral in a solvent) usually involve the application of mechanical energy by high-speed shearing, ultrasonication or high-pressure homogenization; wet and dry methods can be combined together. Chemical treatments may also be applied, which involve (i) the addition of a chemical dispersant to the suspension of the fibrous clay mineral; (ii) chemical functionalization of the palygorskite surface, or (iii) acidification treatment to remove impurities inside the crystal bundles. Ideally, bulk bundles and aggregates of palygorskite should be disrupted into smaller bundles or single rods without damaging the crystal structure and length of nanorods.

Various examples of the aforementioned treatments can be specified. For instance, ultrasound was proposed as an effective treatment to disperse palygorskite in water, with the apparent viscosity rising from ca. 20 to 54 mPa·s [6]. Ultrasonic cavitation causes intense collisions within the palygorskite aggregates promoting its disaggregation. However, under very extreme conditions of ultrasonication, the crystalline structure of palygorskite is disturbed, and palygorskite particles reaggregate [7]. High-speed homogenization was studied by Viseras et al. (1999). According to the authors, stable palygorskite dispersions are difficult to produce by this method, and for a rotor speed of 8000 rpm (10 min), low to medium viscosity gels were obtained [8]. On the other hand, high-pressure homogenization appears to be an efficient approach to disaggregate crystal bundles of palygorskite, as widely reported by Wang and coworkers [3]. The authors found that the bundles could be disaggregated after homogenization at 30 MPa without significant changes of the structure and aspect ratio of the crystals. Too high pressure (90 MPa) afforded a perfect dispersion of the fibers, but it had a negative impact on their length, as shorter fibers appeared [9]. Dispersion of the fibrous clay using this method was used to produce nanocomposites [10].

The addition of chemical dispersants, by changing the charge and surface chemistry of the individual particles (rods) of fibrous minerals, presents itself as an efficient approach to improve and stabilize the dispersions of this type of clay, without damaging the crystal structure. Chemicals such as sodium hexametaphosphate and phosphorus related compounds were used for this purpose [3]. Cetyltrimethylammonium bromide was used to render the palygorskite surface more hydrophobic and enhance dispersibility and compatibility with polymeric matrices [11]. In another study, several quaternary ammonium salts with different lipophilicity were used to coat the palygorskite surface and improve compatibility with an oil-based drilling fluid. A better lipophilicity of the surfactant led to better dispersibility of palygorskite in the oil [12]. The chemical surface functionalization with organosilanes (e.g., 3-glycidoxypropyltrimethoxysilane, 3-aminopropyltriethoxysilane, 3-mercaptopropyltrimethoxysilane, alkyl silanes) was also used to better disperse palygorskite and enhance compatibility with several polymeric matrices [13,14,15].

An efficient disaggregation is crucial for expanding the use of fibrous-like clays, e.g., in composite formulations with cellulose nanofibrils, where the matrix and filler must be homogeneously mixed [16], and in significantly enhancing the performance of the resulting composites [17]. In the present work, two bio-based polyelectrolytes, i.e., sodium carboxymethylcellulose (CMC) and sodium alginate, sodium polyphosphate, and a hydrophobically modified poly(sodium acrylate) (HM-PAA), were evaluated in terms of their efficiency as dispersing agents for palygorskite in water. CMC and alginate were chosen considering their natural and biodegradable origin and their wide availability. Sodium polyphosphate, as mentioned, is commonly used to disperse clay particles in water, and the HM-PAA is an example of a water-soluble synthetic polymer. Additionally, three distinct mechanical mixing systems (magnetic stirring, ultrasonication, and high-speed shearing), and three different pH values were evaluated. With respect to the chemical dispersants here addressed, it was reported the preparation of composites of palygorskite with modified CMC (vinyl grafted) that have superabsorbent properties [18] and with alginate to produce films [19] or foams for the removal of heavy metals [20]. However, a comprehensive study of the effects of the addition of biopolymers/synthetic polymers on the colloidal stability of palygorskite suspensions has not been yet provided.

## 2. Materials and Methods

### 2.1. Clay and Chemical Dispersants

A natural palygorskite sample taken from a deposit located in the department of M’bour, in the region of Thiès (Senegal), south to Dakar, supplied by Tolsa, SA (Madrid, Spain), was used in the present study. The raw material was submitted to preprocessing by micronization in a roller mill to a final particle size >95% smaller than 45 µm, determined by wet sieving, following the method described in API Spec 13A.

The following dispersing agents were used. (i) As bio-based polyelectrolytes, sodium carboxymethylcellulose (CMC) and sodium alginate, both purchased from Sigma-Aldrich (Merck), Algés, Portugal, were included. According to the supplier, the CMC has a molecular weight of 250 kDa and a degree of substitution of 0.7, and the sodium alginate was obtained from brown algae (Bioreagent, catalogue number 71238). No information was provided for the molecular weight of alginate sample, for which we estimated a value of ca. 280 kDa by rheometry measurements. (ii) As synthetic polyelectrolytes, sodium polyphosphate (Emplura grade, catalogue number 106529) purchased from Sigma-Aldrich (Merck) and hydrophobically modified poly(sodium acrylate) (HM-PAA) were used. The polyacrylate with the commercial name Acusol 820, acquired from Rohm and Haas, Philadelphia, PA, USA is a 30% emulsion based on 40% methacrylic acid, 50% ethyl acrylate, and 10% stearyl oxypoly ethyl methacrylate.

All chemicals were used as received without any further purification.

### 2.2. Characterization of the Palygorskite

The sample used in this study was initially characterized for its mineralogical, chemical and physical properties. This characterization comprised the use of X-ray diffraction (XRD), wavelength dispersive X-ray fluorescence (WDXRF) spectrometry, thermal analysis by simultaneous thermogravimetry and differential scanning calorimetry, and Fourier transform infrared–attenuated total reflection (FTIR-ATR) spectroscopy. Additionally, the morphology of the particles was also evaluated by field emission scanning electron microscopy (FE-SEM). The protocols used for the characterization of the palygorskite sample are the same as those described in detail previously for the characterization of sepiolite samples [21].

### 2.3. Preparation of the Palygorskite Suspensions with Different Mechanical Dispersers and Chemical Dispersants

First, aqueous suspensions of 1.0 wt % of palygorskite were prepared at room temperature by adding the desired amount of clay powder into distilled water and stirring at 200 rpm using a magnetic stirrer. After this, the chemical dispersant was added at a concentration of 0.1 wt.%, under constant mixing. For comparison, experiments were also performed without the addition of chemical dispersant. Then, the suspensions were submitted to different mixing systems, namely, a magnetic stirrer at 300 rpm for 20 min, a high-shear disperser (Dispermat CV3-PLUS-E, VMA-Getzman GmbH, Reichshof, Germany) at 5000 rpm for 15 min, or an ultrasound probe (Vibra-cell VC 505, Sonics, Newtown, CT, USA) working at 60% amplitude and 1 s pulse, for 10 min. For some of the experiments, the pH of the suspension was adjusted to either 3 (using HCl 1 M) or to 12 (using NaOH 1 M); the other experiments were performed without any pH adjustment (pH of original clay is ca. 8). Then, the suspensions were left to stabilize for particle size determination, zeta potential measurements, and microscopic analysis. The suspension stability was evaluated for a period of 20 days.

### 2.4. Characterization of the Palygorskite Suspensions

Zeta potential measurements of the palygorskite suspension at different pH values were carried out by electrophoretic light scattering in a Zetasizer Nano ZS ZN 3500 equipment from Malvern (Malvern Instruments, Malvern, UK). An aqueous suspension of the mineral sample (0.5%, *w*/*v*) was first stirred in a high-speed mechanical stirrer (Dispermat CV3-Plus-E) at 3000 rpm for 15 min. The stirring was stopped and the zeta potential of this suspension with a pH of ca. 7.5 was measured. Then, the suspension was placed again under mechanical stirring for 8 min, after which the pH was adjusted either to lower values by the addition of HCl (acidic series) or to higher values by the addition of NaOH (alkaline series), and the zeta potential was immediately measured. This procedure (i.e., mechanical stirring before pH adjustment) was repeated between measurements. Measurements were done by taking the average of six repetitions. It is worth noting that palygorskite suspension in water, without any added dispersants, is not stable. However, high-speed mechanical stirring retards the settling of the palygorskite particles and enables the measurement of their zeta potential. Different series of measurements were performed in order to check the reproducibility of the results.

For the characterization of mineral suspensions, after the treatments referred to in Section 2.3, the palygorskite suspensions at 1 wt % were diluted to 0.1 wt % using MilliQ water and transferred to the measurement cell before the measurements.

To evaluate the size of the palygorskite particles in suspension, dynamic light scattering measurements were performed in the Malvern Zetasizer Nano ZS ZN 3500 equipment (Malvern Instruments, Malvern, UK), with a 532 nm laser, using a backscatter angle of detection of 173°, at 25 °C. Stable palygorskite suspensions with the concentration of 0.1 wt % were gently transferred to a glass cuvette and checked for the presence of bubbles. The particle size was determined from the intensity-weighted distribution (Di50) and based on six repetitions. Data treatment to extract the size distribution and the polydispersity index (PDI) was based on the non-negative least-squares (NNLS) algorithm and was obtained through the Zetasizer Nano Software v7.12.

An Olympus BH-2 KPA microscope (Olympus Optical Co., Ltd., Tokyo, Japan) equipped with a high-resolution CCD Olympus ColorView III color camera was used to evaluate the homogeneity state of the suspensions. Samples were kept between cover slips, illuminated with linearly polarized light and analyzed through a cross polarizer. Microscopic images were captured and analyzed using the analySIS software v5.0 (Soft Imaging System GmbH, Münster, Germany).

The dispersion state of the suspensions, and the morphology and size of the palygorskite particles, after dispersion in aqueous media (as described in Section 2.3), were evaluated also by high resolution transmission electron microscopy (HR-TEM) in a JEOL JEM-2100 instrument (Tokyo, Japan) with a LaB6 filament using an accelerating voltage of 200 kV. A small amount of suspension (10 µL) of each sample was deposited on carbon-coated copper grids for HR-TEM analysis.

### 2.5. Rheology of Aqueous Solutions of the Chemical Dispersants

Mechanical rheometry was used to assess the viscosity of aqueous solutions of the dispersants. The rheological measurements were performed on a controlled stress rheometer (Haake, Model RS1, Karlsruhe, Germany) coupled with cone-plate geometry (C60/1), at a constant temperature of 20 °C. The temperature was controlled using a recirculation bath (Haake Phoenix II, Karlsruhe, Germany). Flow curves of solutions of the different dispersants, at two different concentrations, were obtained in a controlled stress mode with shear stresses ranging between 0.05 and 100.0 Pa, depending on the viscosity of the solutions.

## 3. Results

### 3.1. Characterization of the Palygorskite

The mineralogical composition of the studied palygorskite sample showed the presence of palygorskite as the main clay mineral and a wide range of contaminants, such as quartz, opal-CT (hydrated cristobalite and/or tridymite), calcite, dolomite, and sepiolite. A residual apatite-rich phase (hydroxylapatite-fluorapatite) was also discriminated. In the ~15.0–15.3 Å region, a not well-resolved maximum possibly related with the presence of vestigial Mg,Ca-smectite occurred (Figure 1).

The chemical composition of the studied material is presented in Table 1. The high content of silicon, magnesium, aluminum, and iron evidenced the nature of this sample, which is related to the presence of clay minerals, namely palygorskite (main clay mineral) and sepiolite [Si_12_O_30_Mg_8_(OH)_4_(OH_2_)_4_·8H_2_O] as vestigial clay mineral. The silicon content also comes from quartz (SiO_2_) and opal-CT (SiO_2_·nH_2_O). The calcium can be mostly attributed to calcite (CaCO_3_) and, with the additional presence of magnesium, to dolomite [CaMg(CO_3_)_2_]. A relevant amount of phosphorous (~2 wt %), possibly associated to hydroxylapatite [Ca_5_(PO_4_)_3_(OH)] and/or fluorapatite [Ca_5_(PO_4_)_3_F], was quantified. The high content of iron (~4 wt %) is common in other palygorskites, as identified by Middea et al. (2013) [22].

Appendix A presents the thermogravimetric (TG) curve and corresponding derivative (dTG) of the palygorskite. The first and major endothermic weight loss (~8.0%), detected up to approximately 100 °C, is mainly due to the release of hygroscopic and a portion of zeolitic water from palygoskite (main clay mineral). A second endothermic weight loss (~2.6%), observed from ca. 100 to 225 °C, is attributed to the release of the remaining zeolitic water. The dehydration of the coordinated water occurs in a third stage (~3.7%), extending up to about 500 °C. A final endothermic weight loss step, from ca. 500 to 1000 °C (~4.7%), is related to the dehydroxylation of the hydroxyl groups [23,24,25] and to the decarbonation of calcite [26] and dolomite [27]. The sepiolite (residual clay mineral) has a thermal behaviour similar to that of palygorskite, as stated by Frost and Ding (2003) [23]. The total thermogravimetric weight loss (~19.0%) is in agreement with the loss on ignition value (18.8%) obtained by WDXRF (Table 1).

The palygorskite was additionally characterized by FTIR-ATR spectroscopy (Appendix A). The band at 3612 cm^−1^ is associated to the OH stretching in the Al_2_–OH, Mg_2_–OH, and Al–Fe–OH modes, whereas the band at 3544 cm^−1^ is related to the OH stretching in the Al-Mg–OH and Mg–Fe–OH modes [28]. A diffuse region between 3500 and 3000 cm^−1^ may include bands attributed to adsorbed, zeolitic, and coordinated water. Two close bands related to OH bending, observed at 1658 and 1647 cm^−1^, come from zeolitic and adsorbed water, respectively [29]. A weak and broad band centered at 1432 cm^−1^ is most probably associated with the ν3 (asymmetric stretching) vibration mode of the C–O bonds in carbonate groups, which confirms the presence of calcite/dolomite as contaminants identified by XRD. According to Frost et al. (2001) [30], the bands at 1193, 1086, and 1012 cm^−1^ can be attributed to the Si–O stretching modes, whereas the bands at 974 and 911 cm^−1^ represent the contribution of Mg–OH or Al–OH deformations.

The SEM microphotograph (Figure 2) clearly shows the general fibrous morphology of the studied sample, composed of bundles and individual long rods with a thin diameter. It could also be observed that there are some rhombohedral/prismatic shapes associated with the contaminants present in the sample, such as calcite.

Zeta potential results for 0.5% (*w/v*) aqueous suspensions of clay are shown in Figure 3. At pH 7.5 (initial pH), the zeta potential of the clay suspension was around −17 mV. Typically, zeta potential tended to increase (in absolute value) with the increase of pH. In fact, increasing the pH until 12 increased the zeta potential to −42 mV. However, this variation was not straightforward, and from pH 7.5 to 9.5, zeta potential roughly exhibited a plateau. From the initial pH down to lower pH values, the zeta potential decreased slowly and approximately in a linear manner; at pH 3.5, it was −9 mV, and at pH 1.5 it was −2 mV. The isoelectric point could not be detected within the studied pH range. Although palygorskites from different origins may show different “purity” and surface properties, the present results compare well with other data on palygorskites. Zeta potential values in water in the −15 to −25 mV range have been reported [22,31,32,33,34], being the present value within this range. The absence of detected isoelectric point was also observed for other palygorskite samples [22,35].

### 3.2. Characterization of the Palygorskite Suspensions

Figure 4 shows the macroscopic aspect and the corresponding optical microscopy images of suspensions of palygorskite prepared using different dispersing equipment, in the presence of sodium polyphosphate and without addition of chemical dispersant, at natural pH (ca. 8), for 20 days after sample preparation.

Macroscopically, it was clearly observed that there was an improvement in the suspension stability with the use of a dispersing equipment of higher power, as is the case of the ultrasonicator. The effect of the addition of a dispersing agent, polyphosphate, only resulted in a significant improvement of the macroscopic dispersion stability when the high-speed disperser was used. Additionally, a small decrease of the size of the particles was observed by optical microscopy when the high-speed disperser was used, in comparison to the magnetic stirrer. Using ultrasonication, and in agreement with the macroscopic colloidal stability of the obtained suspensions, the dispersed particles visualized by optical microscopy are apparently smaller than the nonstabilized particles obtained using the high-speed homogenization (Figure 4).

In Table 2, the results obtained for the particle size of the prepared suspensions are summarized. Here, 18 out of 45 preparations generated stable and homogeneous suspensions for 20 days or more after their preparation. Typically, most of the stable suspensions were obtained using ultrasonication, whereas no stable suspensions were obtained using the magnetic stirrer for the sample preparation. With high-speed shearing, a few homogeneous suspensions (i.e., six) could also be obtained. The ultrasonicator was the only method that could provide a high number of stable suspensions (twelve) for different pH values, which indicates its high efficiency for dispersing fibrous clay minerals, as already outlined previously [36]. Regarding pH influence, an increase in the suspension pH led to an increment in the number of stable suspensions, i.e., from 4 suspensions out of the 15 investigated at pH 3, to 8 suspensions out of the 15 investigated at pH 12. It is therefore possible to infer that the dispersions of palygorskite in water present two main tendencies, i.e., the disaggregation and dispersion stability increase (1) as the power of the dispersion equipment is increased, and (2) as the pH is increased. These trends can be rationalized in the following way: (1) The increase in the power of the dispersing equipment results in higher disaggregation of the micro- and nanoparticles, leading to particles of a smaller size and also the formation of more stable suspensions. (2) The rise of the pH causes an increase in the negative surface charge of the clay particles (Table 3), derived from the ionization of hydroxyl groups or breakage of M–O–M bonds (M = Si, Mg, or Al), which are favoured for highly alkaline pH [3]. An increase in the absolute charge value, obtained by pH increment, will be favourable to enhance disaggregation and avoid reaggregation of the particles in suspension, which can result in an enhanced stability of the suspensions.

It was also observed that for high pH (pH 12), using ultrasonication in all cases or even high-speed disperser for most of the cases (experiments with polyphosphate, polyacrylate and CMC) led to homogeneous aqueous phases of palygorskite during the evaluated period of the time (20 days). These results clearly show the major influence of the pH on the stability of the palygorskite suspensions.

Additionally, the particle size of the suspensions tended to decrease with a rise in pH and the polydispersity index decreased as well, indicating a suspension with a narrower distribution of the particle size (Table 2).

Two synthetic and two bio-based dispersing/stabilizing agents were evaluated. To understand whether the viscosity, which can be induced by the presence of those chemicals, is a key factor in the stabilization of the palygorskite suspensions, the dispersants were studied for their viscosity properties in an aqueous solution, in the absence of the mineral. Measurements were made for two different dispersant concentrations and at the three distinct pH values previously addressed (Appendix A). At the concentration of the dispersants used to stabilize the palygorskite suspensions (0.1 wt.%), all the solutions presented very low viscosity, below ca. 18 mPa·s, independently of the pH of the solution, with the exception of alginate at a low pH (pH of 3) due to alginate aggregation and gelation [37]. When increasing the concentration of the polymers (1.0 wt.%), it was possible to observe very different behaviours, depending on the polymer nature and molecular weight. The two bio-based polymers (CMC and alginate) presented similar behaviours and solution viscosities of a similar magnitude for near neutral and alkaline pH (pH of 12), while they were very different from each other at low pH (Appendix A). CMC solutions did not face gelation at pH 3, contrary to alginate, which is probably related to the alginate monomers nature and higher chain rigidity. On the other hand, the synthetic dispersants showed very different behaviour. Polyphosphate is a low molecular weight substance, and thus the viscosity of its solutions is always very low, i.e., close to the solvent (water) viscosity (Appendix A). Polyacrylate behaviour is very dependent on the solution pH (Appendix A). At pH values above ca. 6.5, the polymer is extended, and the solutions containing 1.0 wt % polyacrylate are very viscous, due to the high molecular weight of the polymer and the formation of an entangled polymer network. At low pH values, the polymer collapses, and the solution viscosity is very low, approaching that of water [38].

It was observed that among the studied agents, polyphosphate and CMC enabled the preparation of stable colloidal suspensions in a wider range of conditions. With polyacrylate and alginate, when only using ultrasonication but not at near neutral pH, or with high-speed homogenization in one case each, it was possible to prepare stable suspensions of palygorskite within the established 20-day period. Without the addition of any chemical dispersants, stable suspensions were obtained only when employing ultrasonication, as high-speed homogenization was not effective. These observations indicate that the viscosity of the solutions, induced by the presence of the stabilizers/dispersants, is not the key factor to stabilize the palygorskite suspensions but the interactions between the dispersing/stabilizing agents and the mineral nanocrystals.

To get a further insight into the morphology, aggregation, and size of the palygorskite particles when dispersed at near neutral pH using ultrasonication, TEM images of the suspensions prepared with and without the addition of chemical dispersant were obtained. In Figure 5, significant differences in the micro and submicron particles of palygorskite were observed, as a function of the type of dispersant added. In the micrograph for the suspension prepared without chemical dispersants (Figure 5a), aggregated nanocrystals could be observed, with sizes in the order of 500–800 nm in length and 25–30 nm in thickness. “Face-to-face” aggregated bundles are observed, indicating that the disaggregation was not fully achieved in the absence of a dispersing agent. With the addition of polyphosphate, a better disaggregation was obtained. It was possible to observe single crystals with 500–900 nm in length and 10–20 nm in thickness, as well as some “face-to-face” aggregated crystals (Figure 5b). On the other hand, the use of HM-PAA (Figure 5c) led to a similar disaggregation level of polyphosphate, but the bundles appeared as disordered aggregates, revealing that this dispersing agent is able to disaggregate the crystal bundles of palygorskite. Although due to the nature of this polymer, the suspension was not well stabilized (formation of gel occurred), and aggregation was observed some days after the suspension preparation (Appendix A). This behaviour was also observed when this dispersing agent was used with sepiolite [21]. Conversely, the use of a bio-based polyelectrolyte as a dispersant, i.e., CMC, led to a fully disaggregated suspension (Figure 5d). The nanorods of palygorskite appear as individual rods with 250–600 nm in length and 10–20 nm in thickness. Additionally, some breaking of nanocrystals can be anticipated, with this rupture being induced by the power characteristics of the ultrasonicator operating by cavitation phenomena. The cavitation is responsible for the good efficiency of disaggregation of palygorskite bundles [39]; however, due to its great intensity, it can result in the size reduction of individualized nanocrystals.

CMC and alginate are polymers with some similarities. In addition to both being bio-based, they possess similar pKa values for their carboxylic acid groups (ca. 4.6 for CMC and 4.5 for alginate, respectively [40]), and both are amphiphilic polymers in nature [41,42], enabling different types of interactions between the clay and the biopolymers. Thus, favourable interactions of the biopolymers with the clay particles, derived from their nature, are expected to allow for a better stabilization and dispersion for palygorskite particles in aqueous suspensions. However, as stated in a previous paper [21], alginate possesses a higher degree of substitution by carboxyl groups than the used CMC (higher electrostatic repulsion with the palygorskite fibers), and, additionally, it is based on a more rigid structure (because of the presence of guluronate blocks with an axial–axial conformation). These characteristics limit the flattening of the alginate chains on the palygorskite fibre surface and therefore also its behaviour as dispersing agent for this type of clay. Accordingly, only three successful stable colloidal suspensions of palygorskite were produced when using alginate as a chemical dispersant. Considering the structural characteristics of the biopolymers and palygorskite, the interactions involved are hypothesized to be mainly hydrogen bonding between the hydroxyl groups of the biopolymers and the ionized silanol groups on the outer surface of palygorskite. Water molecules bonded to the external metal atoms in the octahedral sheet (coordinated water) may also interact with the biopolymers. Direct interaction of the carboxyl functional groups from biopolymers with the clay surface may be less favoured, particularly for pH 8 and pH 12, once those groups are negatively charged and the palygorskite surface is, overall, also negatively charged. However, after the eventual biopolymer adsorption on the palygorskite surface, the negative charge of the ionized carboxyl groups could be helpful in restraining the clay–clay aggregation, thus enhancing the dispersion state of the palygorskite particles. In fact, better results were obtained with CMC for higher pH values.

Polyphosphate is traditionally used as a dispersing agent for clay suspensions [43]. This compound presents a very low pKa value, being ionized and very negatively charged in the entire range of studied pH [44]. It was thus expected that it could act as a good dispersing/stabilizing agent even at low pH, even though an excess of negative charge may also limit a good interaction with the also negatively charged clay particles at higher pH values. In fact, the present results obtained indicate a high ability for the dispersion and stabilization of palygorskite particles in an aqueous suspension; this is much improved compared to the experiments without a dispersing agent. Additionally, individualized crystals were obtained at near neutral pH conditions, using ultrasonication. For the HM-PAA, an apparent better performance at pH 12 was observed, where the polymer is extended and can interact with the clay particles; specifically, two homogeneous suspensions using ultrasonication and high-speed homogenization were obtained. This polymer possesses hydrophobic modification, which leads to a shift in the expansion (less expanded) of the polymer chain [38]; this may explain the poor stabilization effect observed for lower pH values.

In summary, it can be stated that the use of ultrasonication is recommended to obtain stable colloidal suspensions of palygorskite, and this usage typically worked for all pH levels and chemical dispersants tested. High-speed shearing may also be used to improve the disaggregation and stability of the suspensions of palygorskite in an aqueous medium, if working with specific dispersants, such as CMC or polyphosphate (both at near neutral pH and at high pH). On the other hand, regarding particle size (Figure 5 and Table 2), for near neutral pH conditions, using CMC or polyphosphate provided a higher degree of stabilization of individualized palygorskite fibers, in comparison to the experiments without a chemical dispersant or with polyacrylate.

## 4. Conclusions

In the present work, a study was conducted that aimed at developing appropriate conditions for the preparation of stable colloidal aqueous suspensions of palygorskite, a mineral whose fibrous particles have a propensity to aggregate in water to form bundles and large aggregates. Most of the palygorskite applications require the mineral to have a good dispersion in water, namely when this mineral is used as a colloidal/stabilizing agent or as a carrier of catalysts, or when used for superabsorbent materials, controlled drug delivery, and tissue engineering, among other uses; this issue is particularly important when dealing with the production of composites. The previous characterization of the palygorskite sample showed it to be composed, aside from palygorskite, of several contaminants, such as quartz, opal-CT, calcite, dolomite, an apatite-rich phase, and sepiolite. Of the three physical treatments employed (i.e., magnetic stirring, high-speed shearing, and ultrasonication), ultrasonication was clearly the most effective procedure in generating homogeneous and stable suspensions, with a low concentration of palygorskite, for at least 20 days after their preparation. Some interesting results could also be obtained using high-speed homogenization, although typically the prepared suspensions were not found to be as stable for the same evaluated period of time as those prepared by ultrasonication; on the other hand, magnetic stirring was completely ineffective. As a tendency, highly alkaline medium (pH = 12) generated more stable suspensions, which is in agreement with the more negative zeta potential value (−42 mV) of the palygorskite sample for this pH. The biopolymer CMC and the polyphosphate were the chemical dispersants that worked better among the tested ones. Overall, from the results of the present study, including particle size assessment by dynamic light scattering and transmission electron microscopy, ultrasonication combined with the addition of CMC or polyphosphate, at near neutral pH, can be recommended for the preparation of suspensions of well-dispersed palygoskite particles in water.

## Figures and Tables

**Figure 1 polymers-13-00129-f001:**
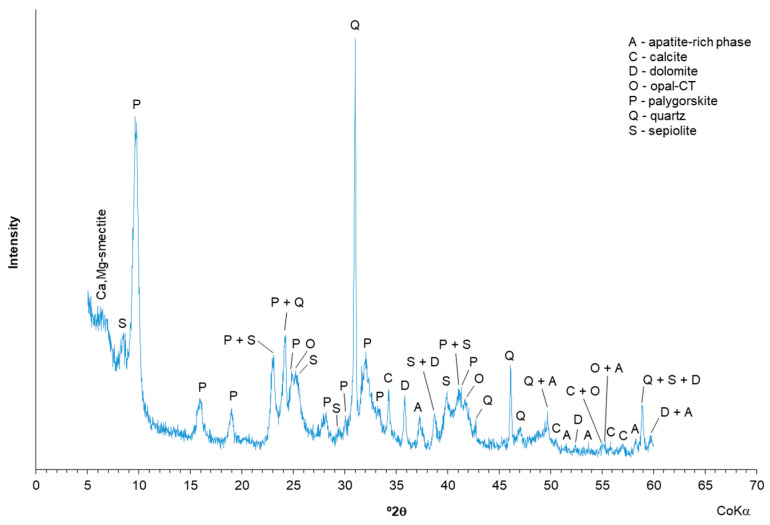
XRD of the palygorskite with identification of the mineral phases.

**Figure 2 polymers-13-00129-f002:**
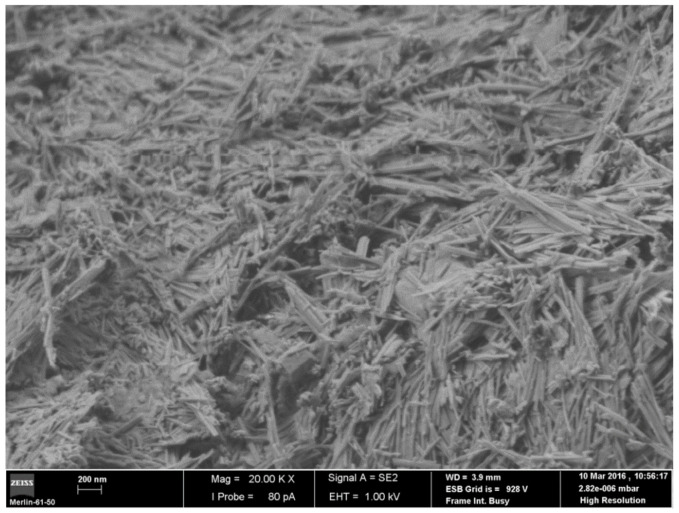
FE-SEM microphotograph of the palygorskite (×20,000).

**Figure 3 polymers-13-00129-f003:**
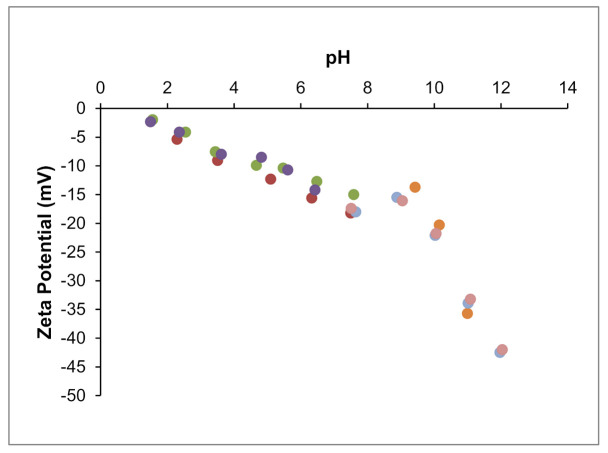
Zeta potential as a function of pH for the palygorskite (different series of measurements were performed, each one shown by a different color).

**Figure 4 polymers-13-00129-f004:**
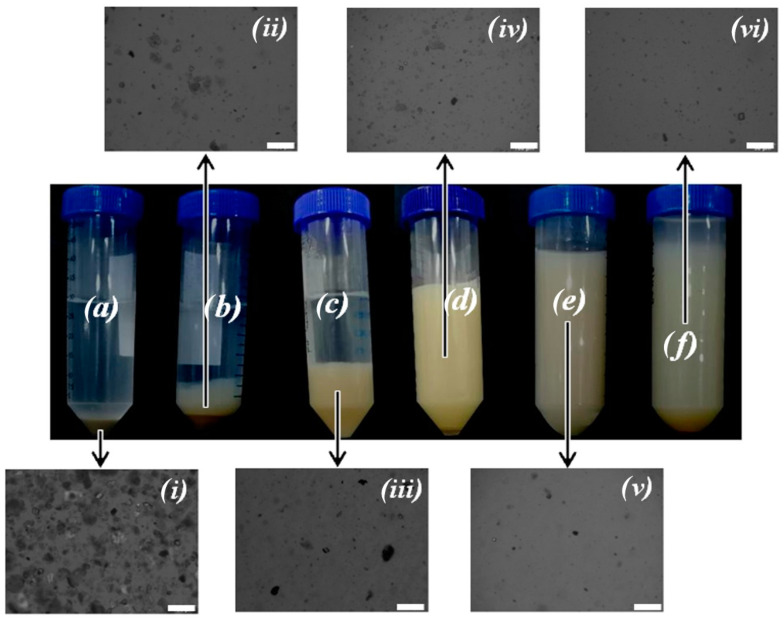
1.0 wt % palygorskite aqueous suspensions (pH ca. 8). (**a**) Magnetically stirred; (**b**) magnetically stirred with 0.1 wt % polyphosphate; (**c**) high-shear dispersed; (**d**) high-shear dispersed with 0.1 wt % polyphosphate; (**e**) sonicated; (**f**) sonicated with 0.1 wt % polyphosphate. (**i**) Bottom phase of the suspension in (**a**); (**ii**) bottom phase of the suspension in (**b**); (**iii**) bottom phase of the suspension in (**c**); (**iv**) the suspension in (**d**); (**v**) the suspension in (**e**); (**vi**) the suspension in (**f**). The images (photographs and micrographs) were taken 20 days after sample preparation. The scale bar in the insets (**i**)–(**vi**) represent 100 µm.

**Figure 5 polymers-13-00129-f005:**
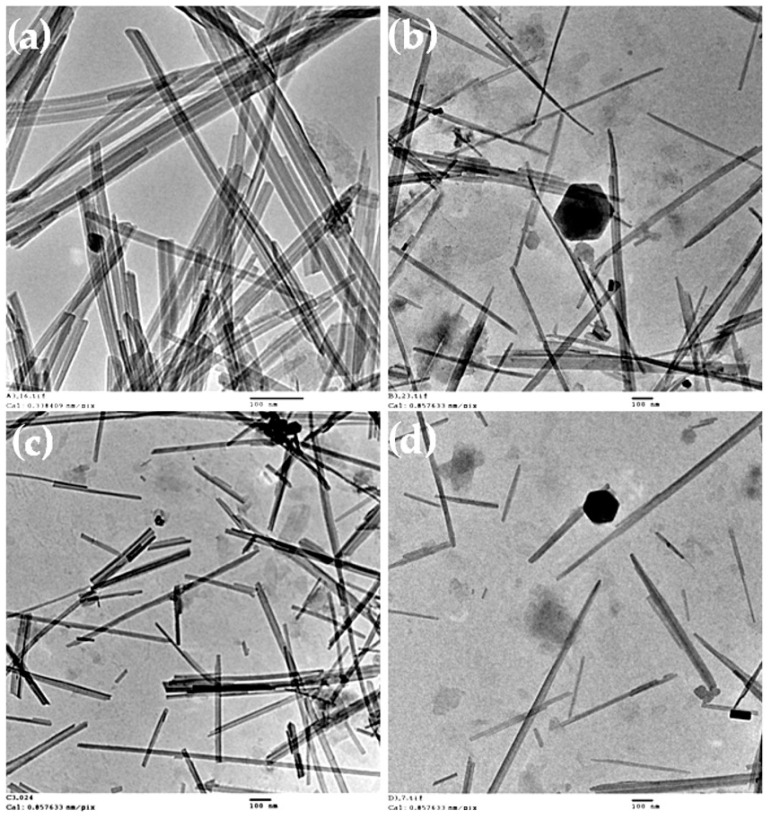
TEM images of 0.05 wt % palygorskite suspensions prepared using the ultrasonic probe at pH ca. 8: (**a**) without dispersing agent, (**b**) with polyphosphate, (**c**) with polyacrylate, and (**d**) with CMC. TEM of the suspension with polyacrylate was done using freshly prepared suspension, due to the high instability of this suspension with time. The scale bars represent 100 nm.

**Table 1 polymers-13-00129-t001:** WDXRF data (in wt %) of the palygorkite.

SiO_2_	53.0
MgO	8.9
Al_2_O_3_	7.0
CaO	4.7
Fe_2_O_3_	4.3
Na_2_O	0.14
K_2_O	0.30
MnO	0.02
TiO_2_	0.34
P_2_O_5_	1.8
SO_3_	0.04
F	0.47
Cl	0.04
Cr	0.06
V	0.03
LOI ^a^	18.8

^a^ LOI: loss on ignition at 1000 °C.

**Table 2 polymers-13-00129-t002:** Particle sizes of palygorskite as a function of the different conditions used for preparation of the suspensions.

	Mechanical Treatment	pH 3		pH 8		pH 12	
		Di50 (nm)	PDI	Di50 (nm)	PDI	Di50 (nm)	PDI
Without dispersing agent	Magnetic stirring	2φ	-	2φ	-	2φ	-
	High-speed homogenization	2φ	-	2φ	-	2φ	-
	Ultrasonication	313	1.00	827	0.71	563	0.30
With 0.1% polyphosphate	Magnetic stirring	2φ	-	2φ	-	2φ	-
	High-speed homogenization	2φ	-	544 *	0.25	286 *	0.26
	Ultrasonication	837	0.96	506	0.27	330	0.25
With 0.1% polyacrylate	Magnetic stirring	2φ	-	2φ	-	2φ	-
	High-speed homogenization	2φ	-	2φ	-	1502	0.48
	Ultrasonication	290	1.00	2φ	-	509	0.27
With 0.1% CMC	Magnetic stirring	2φ	-	2φ	-	2φ	-
	High-speed homogenization	2φ	-	1172 *	0.34	937	0.43
	Ultrasonication	2φ	-	485	0.27	552	0.34
With 0.1% alginate	Magnetic stirring	2φ	-	2φ	-	2φ	-
	High-speed homogenization	2φ	-	1680 *	0.60	2φ	-
	Ultrasonication	659	0.71	2φ	-	242	0.23

2φ—refers to two separate phases; * an aqueous phase with dispersed particles was found together with some amount of solid settled in the bottom of the tube.

**Table 3 polymers-13-00129-t003:** Zeta potential values for palygorskite suspensions as a function of the different conditions used.

	Mechanical Treatment	pH 3		pH 8		pH 12	
		Zeta potential (mV)	Zeta Deviation	Zeta Potential (mV)	Zeta Deviation	Zeta Potential (mV)	Zeta Deviation
Without dispersing agent	Magnetic stirring	2φ	-	2φ	-	2φ	-
	High-speed homogenization	2φ	-	2φ	-	2φ	-
	Ultrasonication	−18	3.0	−20	3.1	−44	4.7
With 0.1% polyphosphate	Magnetic stirring	2φ	-	2φ	-	2φ	-
	High-speed homogenization	2φ	-	−47	3.2	−57	5.9
	Ultrasonication	−24	3.0	−35	4.7	−49	7.6
With 0.1% polyacrylate	Magnetic stirring	2φ	-	2φ	-	2φ	-
	High-speed homogenization	2φ	-	2φ	-	−45	3.7
	Ultrasonication	−16	2.8	2φ	-	−50	5.0
With 0.1% CMC	Magnetic stirring	2φ	-	2φ	-	2φ	-
	High-speed homogenization	2φ	-	−47	3.8	−59	4.2
	Ultrasonication	2φ	-	−53	4.3	−54	4.2
With 0.1% alginate	Magnetic stirring	2φ	-	2φ	-	2φ	-
	High-speed homogenization	2φ	-	−53	3.8	2φ	-
	Ultrasonication	−23	3.7	2φ	-	−49	4.4

2φ—refers to two separate phases; zeta deviation is the standard deviation of the zeta potential measurements.

## Data Availability

The data presented in this study are available on request from the corresponding authors. The data are not publicly available due to lack of adequate repository.

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
