# Peer review of "Stabilization of Palygorskite Aqueous Suspensions Using Bio-Based and Synthetic Polyelectrolytes"

_polymers, 2020, doi:10.3390/polym13010129_

Round 1
Reviewer 1 Report
Ferraz et al. have demonstrated the effect of negatively-charged molecules on the stability of aqueous dispersion of fibrous clay minerals (palygorskite) at different pH values using three different mixing methods. Microscopy methods and dynamic light scattering were used to assess stability. In my opinion, the findings are interesting and generally well explained. I consider it is acceptable for publication after a minor revision.
The authors might consider demonstrating the importance and potential applications of the water-dispersible fibrous clays by adding a couple of sentences in the Conclusion section.
Mention this recent publication in the references:
M. Mar González del Campo, Borja Caja-Munoz, Margarita Darder, Pilar Aranda, Luis Vázquez, Eduardo Ruiz-Hitzky, “Ultrasound-assisted preparation of nanocomposites based on fibrous clay minerals and nanocellulose from microcrystalline cellulose”, Applied Clay Science, 189, 2020, 105538. https://doi.org/10.1016/j.clay.2020.105538.
Section 4 (Discussion) is missing!
Author Response
-Reviewer #1:
Ferraz et al. have demonstrated the effect of negatively-charged molecules on the stability of aqueous dispersion of fibrous clay minerals (palygorskite) at different pH values using three different mixing methods. Microscopy methods and dynamic light scattering were used to assess stability. In my opinion, the findings are interesting and generally well explained. I consider it is acceptable for publication after a minor revision.
The authors might consider demonstrating the importance and potential applications of the water-dispersible fibrous clays by adding a couple of sentences in the Conclusion section.
Mention this recent publication in the references:
- Mar González del Campo, Borja Caja-Munoz, Margarita Darder, Pilar Aranda, Luis Vázquez, Eduardo Ruiz-Hitzky, “Ultrasound-assisted preparation of nanocomposites based on fibrous clay minerals and nanocellulose from microcrystalline cellulose”, Applied Clay Science, 189, 2020, 105538. https://doi.org/10.1016/j.clay.2020.105538.
Section 4 (Discussion) is missing!
Response: Thank you very much for the nice appreciation of our work. Following the reviewer recommendation, we added a sentence in the conclusions section for the interest and potential applications of the water-dispersible fibrous clays, but we should note that these had been already mentioned in the introduction section (second paragraph). See new text added in conclusions section, highlighted in yellow. As for the reference suggested, this was added to the text as well, in section 3.2.
Also, the last part of the manuscript concerning the discussion of the results of dispersion experiments was improved.

Reviewer 2 Report
In the following study the stability of the clay mineral, palygroskite, in aqueous suspension was tested using three different agitation methods and with four chemical stabilizers. For application of palygroskite, achieving high stability and homogenous dispersion is indeed of great importance. In that respect the study is timely and has merit. Still, the research needs to be improved to be suitable for publication. First, the characterization of palygorskite alone is not new and can not stand alone. I understand the extensive characterization serves to explain the trends in aggregation, yet, the trends observed by XRD, SEM, IR and zeta potential measurements have been previously reported (i.e. https://doi.org/10.1016/j.saa.2012.07.085, https://doi.org/10.1016/j.clay.2005.10.005, https://doi.org/10.1016/j.clay.2016.07.012). Second, the choice of the different stabilizers is unexplained, the different treatments seem unorganized and not complete for all the chosen chemicals. This inconsistency makes it difficult to follow the results and to generalize.
We suggest the authors add characterization (IR and zeta (for the samples that were not stable after 20 days)) of the palygorskite with the different stabilizers – to compare to the bare clay and give better insight regarding the results. In addition, the initial characterization should be shortened and partially moved to the supplementary (SEM/zeta and XRD). Lastly, the results section should be revised so the rational for the treatments is better explained.
Author Response
-Reviewer #2:
In the following study the stability of the clay mineral, palygroskite, in aqueous suspension was tested using three different agitation methods and with four chemical stabilizers. For application of palygroskite, achieving high stability and homogenous dispersion is indeed of great importance. In that respect the study is timely and has merit. Still, the research needs to be improved to be suitable for publication. First, the characterization of palygorskite alone is not new and cannot stand alone. I understand the extensive characterization serves to explain the trends in aggregation, yet, the trends observed by XRD, SEM, IR and zeta potential measurements have been previously reported (i.e. https://doi.org/10.1016/j.saa.2012.07.085, https://doi.org/10.1016/j.clay.2005.10.005, https://doi.org/10.1016/j.clay.2016.07.012).
Response: We agree with the reviewer that the characterization of palygorskite is not new. However, we think that we should present it, because palygorskite samples are not all equal and differ one from each other depending on the origin and process of extraction, being therefore important to show its characterization data. Notwithstanding, we now moved some of the figures related to the characterization of palygorskite to supplementary section, as suggested below, leaving in the body of text only the figures that we think are more relevant for the studies of dispersion shown after, which is the main focus of the paper. Last part of discussion of zeta potential results, still in section 3.1, was also shortened.
Second, the choice of the different stabilizers is unexplained, the different treatments seem unorganized and not complete for all the chosen chemicals. This inconsistency makes it difficult to follow the results and to generalize.
Response: In the last paragraph of the introduction section, we now added the reasons for the choice of the chemical dispersants used in the present work: “CMC and alginate were chosen considering their natural and biodegradable origin and their wide availability. Sodium polyphosphate, as mentioned, is commonly used to disperse clay particles in water, and the HM-PAA is an example of a water-soluble synthetic polymer”.
With regard to the different treatments used, section 2.3 (in the Materials and Methods) and Tables 2 and 3 show that three different mixing apparatus were considered (magnetic stirring, high-speed homogenization, and ultrasonication), three different pHs (pH 3, pH 7.5, and pH 12), and four chemical dispersants (polyphosphate, polyacrylate, CMC, and alginate), being also conducted experiments without the addition of chemical dispersant, which gives a total of 45 experiments. For the cases where homogeneous aqueous suspensions of palygorskite could be obtained for the established period of time after preparation (20 days), the particle size and zeta potential were measured, being the results shown in Tables 2 and 3, and discussed for the main trends along the section 3.2.
We suggest the authors add characterization (IR and zeta (for the samples that were not stable after 20 days)) of the palygorskite with the different stabilizers – to compare to the bare clay and give better insight regarding the results. In addition, the initial characterization should be shortened and partially moved to the supplementary (SEM/zeta and XRD). Lastly, the results section should be revised so the rational for the treatments is better explained.
Response:
The idea of the present work is not the analysis of the samples that are not stable after 20 days. For these samples, the palygorskite solid settled in the bottom of the tube with a clear separation between a water phase (with no solid) and the settled solid, which has no interest for an application. What is intended is the opposite, that is, the dispersion of palygorskite particles in water to provide stable colloidal suspensions. Thus, making FTIR of the solid settled in the bottom of these tubes has, in our opinion, no interest to the present study. This also applies to zeta potential measurements, because, for the not stable suspensions, typically, the liquid phase is only water with the solid settled in the bottom.
As mentioned above, we shortened slightly the part where the characterization of palygorskite was presented and discussed, and moved some related figures to the supplementary section.
Finally, the last part of the manuscript concerning the discussion of the results of dispersion experiments was improved. See new text added in section 3.2 (Lines 387-397), highlighted in yellow. Additionally, we performed a study of the viscosity of aqueous solutions of all the chemical dispersants used in this study, as suggested by the editor. Note that viscosity could be a key factor influencing the stability of the palygorskite aqueous suspensions. These results have been now added to the manuscript and a discussion of it was provided as well (section 3.2).
Round 2
Reviewer 2 Report
The authors improved the manuscript greatly. I agree the viscosity measurments add to the overall understanding. I suggest some editing - mainly, the authors have a tendency to write very long elaborate sentences that are hard to follow. See for example lines 318-322.
Author Response
Reviewer 2#
The authors improved the manuscript greatly. I agree the viscosity measurments add to the overall understanding. I suggest some editing - mainly, the authors have a tendency to write very long elaborate sentences that are hard to follow. See for example lines 318-322.
Reply: We are grateful to the reviewer for recognizing the improvements made in our manuscript. According to the reviewer's suggestion, the elaborated and long sentences were changed. The changes are highlighted in yellow.
